# Dissipation Analysis Methods and Q-Enhancement Strategies in Piezoelectric MEMS Laterally Vibrating Resonators: A Review

**DOI:** 10.3390/s20174978

**Published:** 2020-09-02

**Authors:** Cheng Tu, Joshua E.-Y. Lee, Xiao-Sheng Zhang

**Affiliations:** 1School of Electronic Science and Engineering, University of Electronic Science and Technology of China, Chengdu 611731, China; ctu@uestc.edu.cn; 2Department of Electrical Engineering, City University of Hong Kong, Kowloon, Hong Kong; josh.lee@cityu.edu.hk; 3State Key Laboratory of Terahertz and Millimeter Waves, City University of Hong Kong, Kowloon, Hong Kong

**Keywords:** micro-electromechanical systems (MEMS), piezoelectric MEMS resonators, laterally vibrating resonators, quality factors, dissipation mechanisms, Q-enhancement strategies

## Abstract

Over the last two decades, piezoelectric resonant sensors based on micro-electromechanical systems (MEMS) technologies have been extensively studied as such sensors offer several unique benefits, such as small form factor, high sensitivity, low noise performance and fabrication compatibility with mainstream integrated circuit technologies. One key challenge for piezoelectric MEMS resonant sensors is enhancing their quality factors (*Qs*) to improve the resolution of these resonant sensors. Apart from sensing applications, large values of *Qs* are also demanded when using piezoelectric MEMS resonators to build high-frequency oscillators and radio frequency (RF) filters due to the fact that high-Q MEMS resonators favor lowering close-to-carrier phase noise in oscillators and sharpening roll-off characteristics in RF filters. Pursuant to boosting *Q*, it is essential to elucidate the dominant dissipation mechanisms that set the *Q* of the resonator. Based upon these insights on dissipation, Q-enhancement strategies can then be designed to target and suppress the identified dominant losses. This paper provides a comprehensive review of the substantial progress that has been made during the last two decades for dissipation analysis methods and Q-enhancement strategies of piezoelectric MEMS laterally vibrating resonators.

## 1. Introduction

Resonant sensors, characterized by a resonant frequency dependent on the physical measurand, offer a great advantage over conventional analog sensors in that a frequency outputs can be digitized through mature frequency counting techniques realizable in digital systems [1]. This greatly reduces the complexity and cost generally associated with using the analog-to-digital converters to measure the amplitude of analogue voltages. As the information of the measurand is carried by frequency instead of amplitude, resonant sensors usually exhibit better immunity to interference or noise compared to their analog counterparts [2]. One of the widely-used resonant sensors is the quartz crystal microbalance (QCM), the operation of which relies on direct and inverse piezoelectric effects inherent to quartz resonators [3]. Although QCMs have high Q, large power handling capability and excellent temperature stability for certain cut angles, miniaturization of quartz resonators is still very challenging, which makes it difficult to integrate the quartz resonators monolithically with integrated circuits (IC) [4]. With the rapid development of micro-fabrication techniques, resonant sensors based on various micromachined piezoelectric resonators, such as thin film bulk acoustic wave resonators (FBARs) [5,6,7], laterally vibrating resonators (LVRs) [8,9,10,11] and flexural-mode beam resonators [12,13,14], have been proposed with fabrication processes compatible with mainstream IC technologies. For FBARs and flexural-mode beam resonators, the sensitivity per unit area and resonant frequency are both dependent on the thickness of resonant structures, which makes it hard to decouple these two important design parameters [15]. Moreover, FBARs normally have resonant frequencies in the GHz range due to difficulty in depositing thick piezoelectric films with good quality. It should be noted that a higher operating frequency of a resonant sensor does not necessarily translate into improved overall sensor performance as the noise level usually deteriorates with frequency, and it is generally accepted that higher operating frequency results in larger power consumption and design complexity for the read-out electronic circuit [15]. In contrast to FBARs and flexural beam resonators, LVRs feature lithographically definable resonant frequencies that are independent of thickness. Therefore, the sensitivity of resonant sensors based on LVRs can be improved by reducing the thickness while keeping resonant frequencies unchanged, unlike FBARs. This unique characteristic, combined with compatible manufacturing process with ICs, has rendered LVR a viable candidate for future integrated resonant sensors with miniaturized dimensions, high performance and low power consumption. Over the last two decades, a significant number of high-performance resonant sensors based on LVRs have been demonstrated, such as chemical sensors [16,17], thermal detectors [9,18], infrared sensors [19], biological sensors [20,21], inertial sensors [22,23], magnetometers [24,25] and miniaturized acoustic antennas [26].

In piezoelectric MEMS LVRs, the electric field is mainly applied across the thickness of the piezoelectric film to generate a lateral strain within the plane of the device with suspended free edges. The amplitude of lateral vibration becomes maximum when the frequency of the excitation signal coincides with the mechanical resonance of the resonator. For further details on the working principle of piezoelectric LVRs, we refer readers to references [15,27]. There are two types of electrode configurations that are commonly adopted for LVRs. The first type uses interdigital transducer (IDT) electrodes only on one side of piezoelectric film [28,29], while the second type has top and bottom electrodes on both sides of piezoelectric film [30,31]. Depending on the structure of resonators, piezoelectric MEMS LVRs can also be categorized into two types. The first type of LVRs consists of a piezoelectric film, both as the transducer and acoustic cavity [32]. The commonly-used piezoelectric materials include aluminum nitride (AlN) [32], lithium niobate (LN) [28], zinc oxide (ZnO) [27], gallium nitride (GaN) [33] and lead zirconate titanate (PZT) [34]. The second type of LVRs uses the thin piezoelectric film as a transducer only while having most of the acoustic energy propagating in the underlying thicker substrate [27]. Such resonators are usually referred to as thin-film piezoelectric-on-substrate (TPoS) resonators. Compared to the LVRs using only the piezoelectric film as the resonator body, TPoS LVRs generally exhibit higher Q and larger power handling capability when appropriate substrate materials are adopted. The commonly-used substrate materials include silicon [35], diamond [36] and silicon carbide [37]. As a trade-off, the effective electromechanical coupling (*k_eff_*^2^) of TPoS LVRs is usually lower than piezoelectric-only LVRs as a significant part of the electrical energy transformed into acoustic energy is stored in the substrate layer. To improve the *k_eff_*^2^ of TPoS, a novel type of sidewall-excited LVRs has been reported as a potential solution [38].

Figure 1a shows a typical structure of a TPoS resonator, which consists of input/output IDT electrodes, piezoelectric film, bottom electrode and substrate layer. When Si is used as the substrate layer, a bottom electrode layer could be omitted by highly doping the top surface of Si substrate and let it serve as the “bottom electrode” [39]. The applied AC electric field across the thickness of the piezoelectric film (denoted as z direction) causes it to expand and contract in the y direction through the piezoelectric coefficient d_31_. Figure 1b,c depicts the perspective and cross-sectional views of the indented fundamental symmetrical (S_0_) mode of Lamb wave. It should be noted that the electromechanical transduction efficiency is maximized when the electric field and strain field in the piezoelectric film are matched. That is to say, the indented S_0_ mode can be efficiently excited when the center-to-center pitch of the electrodes equals to half wavelength (*λ*) of the Lamb wave as shown in Figure 1c. As such, the resonant frequency of the LVR is given by:(1)$$f_{0}=\frac{V_{phase}}{\lambda }\approx \frac{1}{2W_{pitch}}\sqrt{\frac{E_{s}}{\rho_{s}}}$$
where *V_phase_* denotes the acoustic phase velocity of the resonator. *E_s_* and *ρ_s_* are the Young’s modulus and mass density of the substrate layer, respectively. It should be noted that Equation (1) approximates the acoustic phase velocity using only the material properties of the substrate layer, which is usually much thicker than the other constituent layers of TPoS LVRs. Equation (1) indicates a key feature of LVRs, which is that the resonant frequency of the device can be set by the lithographically defined dimension. Although Equation (1) uses variable “*W_pitch_*” for the case where IDT electrodes are adopted, one can simply replace *W_pitch_* with other physical dimensions that represent half of acoustic wavelength in LVRs using patch electrodes (e.g., the width of the resonator in the width-extensional LVRs [39]). Figure 1d shows the electrical characterization setup for a 2-port LVR, from which the network parameters such as *Z*, *Y* and *S* parameters can be obtained. A typical plot of the associated transfer admittance *Y*_21_ is shown in Figure 1e, where it can be seen that the magnitude of *Y*_21_ is maximum at the resonant frequency *f*_0_.

The operation of resonant sensors hinges on the shift in resonant frequency due to a change in effective mass or stiffness of the resonant structure when subjected to a perturbation from the quantity of interest. There are three methods that are commonly used to detect the frequency shift of LVRs. The first utilizes the electrical characterization setup shown in Figure 1d, where the frequency response of the resonator is monitored by a network analyzer [10]. The second method involves detecting a change in amplitude of the output signal by exciting the resonator with an AC signal at a fixed frequency around the resonant frequency. This method essentially converts the shift in resonant frequency to a change in amplitude which is usually monitored using a lock-in amplifier [40]. The third method involves building an oscillator by connecting the resonator with the electronic amplifier to form a feedback loop. The output frequency of the oscillator can be readily monitored by a frequency counter [24]. Compared to the first two methods, the third method is more suitable for commercial applications as it does not require cumbersome characterization equipment [41]. It should be noted that, in resonant sensors characterized by all three methods mentioned above, higher *Q* is generally desired for better resolution [4]. Thus, it is of vital importance to maximize the *Q* of LVRs for sensing applications. In addition, boosting *Q* also lowers close-to-carrier phase noise in the application of oscillators [42], and improves roll-off characteristics in the application of RF filters [43].

This paper is organized as follows: Section 2 of this paper introduces different dissipation sources in piezoelectric LVRs, with particular attention on anchor loss and thermoelastic damping (TED), both of which are generally accepted to be dominant in setting *Q* of LVRs. Section 3 reviews the recent progress on numerical and experimental methods for analyzing these two dominant dissipation mechanisms. In Section 4, a variety of Q-enhancement strategies for reducing anchor loss or TED are presented and compared. Section 5 discusses the issues that need to be addressed in relation to existing dissipation analysis methods and Q-enhancement strategies.

## 2. Dominant Dissipations in LVRs

There are many dissipation mechanisms reported for LVRs, which are usually divided into two groups, namely intrinsic and extrinsic loss. Intrinsic losses are fundamental dissipative processes, thus dependent on the material properties, mode of resonance and internal structure. In contrast, extrinsic losses stem from interactions between the structure and the environment. Generally, the intrinsic losses, such as phonon-phonon or phonon-electron interactions, set the maximum *Q* achievable in bulk mode resonators [44]. Thermoelastic damping (TED) is one example of a type of intrinsic loss [45]. In most of the cases, the intrinsic losses are much smaller compared to extrinsic losses and easily get masked by the latter. Extrinsic losses mainly include anchor loss [46], ohmic loss, viscous loss [47] and surface loss [48], each of which will be detailed in the following part of this paper.

Quality factor (*Q*) is defined as the ratio of the maximum energy stored in the resonant system over the energy dissipated per cycle. The overall quality factor (*Q_total_*) of LVRs can be expressed by summing up the distinct dissipations [49]:(2)$$Q_{total}=2\pi \frac{E_{stored}}{E_{dissipated}}={\left(\frac{1}{Q_{anchor}}+\frac{1}{Q_{TED}}+\frac{1}{Q_{Other}}\right)}^{-1}$$
where *Q_anchor_*, *Q_TED_* and *Q_other_* correspond to anchor loss, TED, and other losses, respectively. Unfortunately, there is scarcely any reported work on measuring individual loss (or individual *Q*) directly. Instead, only *Q_total_*, which describes the combined effect of different damping sources, can be obtained by measuring the amplitude-frequency or phase-frequency responses of the resonators. This brings about a huge challenge in analyzing the damping mechanisms present in LVRs because each constituent source of damping cannot be easily analyzed separately. However, if the dominant losses are much larger than other losses, then the other losses may be neglected in an approximation of *Q_total_*. In other words, the approximation considers only the dominant losses, which exhibit lower *Qs* and thus have more significant impact on *Q_total_* as indicated by Equation (2). It should be noted that Equation (2) emphasizes *Q_anchor_* and *Q_TED_* because anchor loss and TED were found to play dominant roles in setting *Q_total_* in many LVRs [46,50,51,52]. Thus, this work mainly focuses on these two dissipation sources. However, other dissipations could become dominant in certain cases, which will be detailed in the following sections.

### 2.1. Anchor Loss

Generally, LVRs consist of free-standing resonant bodies that require mechanical support (dubbed as anchor). Thus, an anchor is defined as a mechanical structure that attaches the resonator body to the peripheral supporting frame. Anchor loss occurs when elastic energy propagates from the resonator body to the surrounding substrate through the anchors. If the total elastic energy stored in the resonator is given by *E_r_* and the energy propagating out via the anchors is given by *E_l_*, then *Q_anchor_* can be computed using the following equation [53]:(3)$$Q_{anchor}=2\pi \frac{E_{r}}{E_{l}}$$

For any LVR with a free-standing resonator body which is suspended by the supporting structures, anchor loss is unavoidable. This is true even if the anchor structure is placed at the positions where minimum displacement occurs (such points are normally referred to as nodal points). This is because the anchor structures have a finite size in practical cases rather than being an idealized infinitesimal point. It is widely reported that anchor loss plays the major role in setting the *Q* in most of LVRs [31,46]. As such, a variety of Q-enhancement strategies that aim to reduce anchor loss have been proposed over the last two decades, some of which will be reviewed and discussed in detail in Section 4.

### 2.2. Electrode-Related Loss and Thermoelastic Damping (TED)

For LVRs utilizing piezoelectric transduction, metal electrodes are necessary for actuation and detection of the Lamb wave in the resonator body. It was found there exists a damping mechanism related to the electrodes that has been associated with interfacial loss [54]. This electrode-related loss was confirmed by a demonstration in a 3.2-GHz overmoded LVR that the *f*·*Q* product can be pushed to 1.17 × 10^13^ by reducing the electrodes coverage to 0.57% [55]. To completely eliminate the electrode-related loss, some researchers proposed a capacitive-piezo transducer, which separates the electrodes from the resonator body by sub-micro gaps [56,57]. Applying this design strategy to a 940-MHz LVR leads to a 5× enhancement in *Q*, resulting an *f*·*Q* product 4.72 × 10^12^. These experimental results suggest that anchor loss alone cannot account for the measured *Qs* in LVRs, and the electrode-related loss also plays a significant role in setting *Q*. However, there is yet to be a solid universal explanation for the electrode-related loss observed in various types of LVRs. The complexity in analyzing electrode-related loss arises due to the uncertainty of the thermal properties and interfacial conditions for constituent metal and piezoelectric thin films [31,54]. One possible cause for the electrode-related loss is TED, which has been extensively studied for micro-resonators using electrostatic transduction [58,59,60,61,62]. For micro-resonators using piezoelectric transduction, it has been experimentally verified that TED plays a dominant role in setting *Qs* of a 1-GHz AlN LVR [51]. TED was also analyzed in a series of 140-MHz AlN-on-Si LVRs using finite-element analysis [50], where it was shown that TED made comparable contributions as anchor loss in setting the *Qs* of regular flat-edge LVRs. Thus, it is reasonable to consider TED as one of the significant source for electrode-related loss.

TED refers to an irreversible energy transfer process as elastic energy turns into heat [63,64]. The dissipation is caused by thermal gradients arising from the local volumetric change as the resonators vibrate. As such, this dissipation mechanism is absent for pure shear modes, where the volume of such modes does not change. This is the reason why reported *f*·*Q* products for Lamé bulk-mode silicon resonators can reach close to the limit set by intrinsic material loss [65]. In comparison, TED becomes more relevant for Lamb wave modes where longitudinal waves and shear waves are usually coupled in the resonator body. It is reported that *Q_TED_* can be evaluated using the following equation [63]:(4)$$Q_{TED}=\frac{c_{v}{}^{2}}{\psi \alpha^{2}\rho kT\omega }$$
where *c_v_* denotes heat capacity per unit volume, *ψ* a constant related to the geometry of the resonator, *α* thermal expansion coefficient, *ρ* mass density, *k* thermal conductivity, *T* absolute temperature and *ω* angular frequency. Although Equation (4) holds for single-material devices, it can be extended to predict *Q_TED_* of composite resonators (e.g., TPoS LVRs) by considering TED from each constituent layer with the help of finite-element analysis tools [50]. It was also found that TED from metals far exceeds that from semiconductors and dielectrics since metals generally exhibit larger thermal conductivities and thermal expansion coefficients [51,66]. Thus, it is reasonable to attribute TED to metal electrodes and simply ignore TED from piezoelectric and substrate layers as a reasonable approximation of *Q_total_*.

### 2.3. Other Dissipation Sources

Apart from anchor loss and TED, other dissipations such as ohmic loss and viscous loss could also become dominant for LVRs in certain cases.

Ohmic loss occurs when electrical currents pass through electrodes with low but non-zero resistance. As the operation frequency of LVRs increases, ohmic losses usually become more significant due to the smaller electrode width required for excitation of high-frequency Lamb waves. Larger electrical loss results in more heat generation causing non-linearity issues in LVRs which further limits the power handing capability [67]. To reduce ohmic loss, a common method is to increase the thickness the metal electrodes. However, a thick metal layer could negatively impact the indented vibration mode or cause additional damping [51]. An alternative solution is to use low-resistivity metals. It should be noted that trade-offs usually exist for choosing the metal materials for LVRs as many other design factors, such as acoustic impedance, power durability and complexity in terms of microfabrication process, also need to be taken into account.

Viscous loss occurs when a portion of kinetic energy transfers from the resonator to the surrounding molecules. It usually becomes larger as the surface to volume ratio of the resonator increases [47]. When operating LVRs in a fluidic medium, which is usually the case for biological sensing applications, viscous loss becomes the most dominant damping mechanism [21]. A common method to reduce viscous loss is operating the resonator in a shear mode as the shear acoustic wave does not displace the molecules perpendicular to the resonator surface [68].

Another dissipation mechanism that also heavily depends on the surface to volume ratio of the resonator body is surface loss. Surface losses are usually associated with the surface non-idealities such as surface roughness, lattice defects, adsorbents and contamination, which could be critical to the performance of resonators when the size of devices shrinks to nano-scale [69,70]. Although the physics of surface losses are complicated, there has been an impressive effort to analyze this damping mechanism using a unified model [48]. It has also been reported that vacuum annealing can be used to significantly reduce surface losses [71,72]. Given the growing research interest in nano-electromechanical LVRs in recent years [11,19,26,73,74], it can be envisioned that more attention will be drawn to study surface losses in LVRs with nano-scale dimensions.

It has been reported that the damping mechanisms related to phonon-phonon and phonon-electron interactions limit the *Q* of bulk (or contour) mode resonators [44,75]. But these losses have been shown to be negligible in LVRs as the experimentally demonstrated *f*·*Q* products are at least an order of magnitude smaller than the theoretically predicted limit [55,76]. In addition, it has been proposed that charge redistribution loss could set the *Q* limit in piezoelectric resonators that operate at frequencies less than 1 GHz [77]. However, there is a scarcity of published material regarding this loss mechanism. As such, these damping mechanisms are not covered in this review.

## 3. Dissipation Analysis Methods for LVRs

As anchor loss and TED are the most dominant loss mechanisms for LVRs, several methods of analysis targeting these two losses have been proposed during the last decade. These methods can be mainly categorized into two types, namely numerical analysis and experimental investigation methods, which can complement each other. This section will provide a comprehensive review on these two types of methods for analyzing anchor loss and TED.

### 3.1. Numerical and Experimental Analysis Methods for Anchor Loss

Anchor loss captures the radiation of elastic energy from the resonator body to surrounding substrate via the supporting structures. The analysis complexity lies in that a portion of elastic waves could be reflected back into the resonator body if the surrounding substrate is finite in size or there exists an acoustic impedance mismatch along the path of outward radiation. In such case, one needs to take the whole substrate in which resonator resides into account, which substantially increases the complexity of analysis. One commonly employed approach to this problem has been to consider the substrate as a semi-infinite domain where all outgoing elastic waves from the resonators are absorbed and no reflection occurs [78]. This approximation is reasonable considering the resonators are usually much smaller compared to the substrate. To mimic the semi-infinite substrate, perfectly matched layers (PMLs) were usually adopted. The concept of PMLs was first introduced to absorb outgoing electromagnetic waves to mimic infinitely large domains when solving problems of electromagnetic wave propagation [79]. The ideal impedance matching characteristics of the PML is realized by coordinate transformation which force the displacements to vanish [80]. Thus, truncating the semi-infinite substrate with a three-dimensional PML with finite size allows the substrate to absorb the outgoing elastic waves from any angle. Given that all the elastic energy arriving at the PML is completely dissipated, the PML technique also facilitates computing the arriving elastic energy from the anchors and thus *Q_anchor_* according to Equation (3) [81].

As most of LVRs adopt straight beam tethers as the supporting structures connecting the resonator body with the substrate, it is meaningful to investigate the simplest case where a cantilever is attached to the substrate. Frangi et al. [80] proposed the implementation of PML for a two-dimensional (2D) cantilever, as shown in Figure 2a. The length and width of the cantilever are L and H, respectively. As the cantilever vibrates, a portion of elastic wave propagates to the semi-infinite substrate on the left causing anchor loss. To mimic the semi-infinite substrate, a Cartesian PML with width of W_PML_ is used to truncate the substrate at certain distance (W_∞_) from the cantilever. The elastic waves radiate from the anchor of the cantilever to the substrate is completely dissipated in PML, which further allows computation of the elastic energy dissipated over one cycle. It should be noted that the computed elastic energy dissipated in PML heavily depends on the physical dimensions of PML such as W_∞_ and W_PML_. Setting the value of W_PML_ too small will cause a reflected wave that perturbs the solution of the wave propagation in the substrate [78]. Specific criteria for the selection of PML parameters is discussed in detail in [80]. To evaluate the anchor loss more accurately, the implementation of PML for the three-dimensional (3D) cantilever is necessary. Figure 2b shows the case where a 3D semi-infinite substrate is truncated by the PML. An alternative to implement PMLs is shown in Figure 2c, where a semi-infinite plate substrate with finite width (W) is truncated by the PML. In this case, besides W_∞_ and W_PML_, it is also important to determine the width of the plate to obtain reliable predictions for *Q_anchor_*. It can be seen from Figure 2d that the computed values of *Q_anchor_* tends to converge when W is sufficiently large. This suggests there exists a minimum value for the plate width for reliable predictions for *Q_anchor_*. As the proposed numerical method is based on finite-element analysis, the reliability of computed *Q_anchor_* needs to be confirmed by refining the mesh of the model, as shown in Figure 2e.

Based on the numerical analysis method using PML, Siddiqi et al. [50] proposed a 3D model to evaluate the anchor loss in a AlN-on-Si resonator (a typical type of TPoS LVR), as shown in Figure 3a. It can be seen from Figure 3a that, by utilizing the symmetry of the resonator, only a quarter section of the device is analyzed with the aim to reduce computation time. The physical dimensions of the substrate, PML and mesh element size should be set in relation to the acoustic wavelength (*λ*) of the Lamb wave propagating in the resonator. Figure 3b shows the simulated vibration mode shape of the intended S_0_ mode, where the acoustic wavelength of 60 μm is labelled. The side length of the square substrate region is set to be about twice the acoustic wavelength, and the length of PML is set to be equal to one acoustic wavelength. The convergence test for the computed *Q_anchor_* was performed over PML length and number of mesh elements, as shown in Figure 3c,d. Figure 3e shows a scanning electron micrograph (SEM) for the tether part of the fabricated resonator. With the help of the numerical analysis method using PML, the values of *Q_anchor_* were computed, which showed good agreement with the experimental results. Furthermore, the proposed analysis method can be used to predict the *Q_anchor_* as functions of tether length, tether width and fillet radius as shown in Figure 3f–h. The work shown in [50] proves that the numerical analysis method based on PMLs is a powerful means to evaluate anchor loss which is expected to significantly benefit the design of high-Q LVRs.

Damping in MEMS resonators are usually characterized by *Q*, which requires measurement of electrical parameters such as *S*, *Z* or *Y* parameters. The electrical characterization methods are convenient and efficient in measuring *Q*, but the measured *Q* does not provide any insight of the nature of damping. Recently, Gibson et al. [82] investigated anchor loss using laser Doppler vibrometry (LDV). It is shown in the work that the undercutting region of a 220-MHz AlN LVR had an out-of-plane displacement as large as 60 pm, which served as a proof that the device release distance had a significant effect in determining the anchor loss. Tu et al. [83] also applied LDV technique to probe the anchor loss in AlN-on-Si LVRs, which showed a strong correlation between the measured *Q* and the out-of-plane displacement profiles on the top surface of the resonator. In this work, two types of resonator, namely flat-edge and biconvex resonators, were investigated by both LDV and electrical characterization methods. Figure 4a shows the SEM of a flat-edge resonator (F80). The flat-edge resonator has a rectangular body with the interdigital electrodes placed on top, which is commonly adopted configuration for LVRs. Figure 4b shows the SEM of the proposed biconvex resonator (B80), the edge of which is curved for confinement of acoustic energy into the center of the resonator and thus reduce anchor loss. The resonant frequency of the device is determined by the electrode pitch (W_p_) which is equal to half of the acoustic wavelength (*λ*). Figure 4c depicts the schematic of a flat-edge resonator showing the constituent layers of AlN-on-Si structure. The constituent layers are the same for both flat-edge and biconvex resonators. The measured *z*-direction out-of-plane displacement profiles by LDV over four devices (F80, B80, F60 and B60) are shown in Figure 4d–g, respectively. It can be seen from Figure 4d,f that maximum displacement with opposite signs occurs alternately along the width of the flat-edge resonators (F80 and F60). In the length direction, the displacement profile is generally uniform, which allows z-direction motions to be transferred to the supporting tethers resulting in anchor loss. In contrast, z-direction displacements are more concentrated in the center for resonators B80 and B60, as shown in Figure 4e–g. Intuitively, these measured results from LDV leads to a hypothesis that biconvex resonators suffer less anchor loss compared to conventional flat-edge resonators since less acoustic energy is leaked to the surrounding substrate via supporting tethers in the latter case. This hypothesis is verified by the measured magnitude of admittance frequency response from the electrical characterization method as shown in Figure 4h, which shows that the measured *Q* of resonator B60 is about 16 times that of F60. This strong correlation between the z-direction displacement profile and measured *Q* is consistently observed for resonators with different resonant frequencies as shown in Figure 4i. The results shown in [83] validates that anchor loss can be probed by visualizing the out-of-plane displacement profile on the top surface of LVRs using LDV.

### 3.2. Numerical and Experimental Analysis Methods for TED

The concept of TED was first proposed by Zener et al. in 1937 [84], where the thermoelastic coupling between acoustic and thermal phonons was studied analytically for flexural vibration modes in thin rods. Although the Zener’s work aims at the fundamental mode of vibration in a beam structure, its implication with regards to the inverse relation between *Q_TED_* and ambient temperature applies generally for MEMS resonators of different shapes [85,86,87,88,89]. Much work has been done to improve the accuracy of evaluating *Q_TED_* in the context of MEMS resonators. Lifshitz et al. [90] proposed an exact analytical expression of *Q_TED_* in thin rectangular beams undergoing small flexural vibrations. To accommodate more general cases other than simple beams, Ardito et al. [91] developed an ad-hoc numerical procedure which was implemented in a finite element program. Segovia-Fernandez et al. [51] proposed a semi-analytical approach that enables computation of *Q_TED_* in LVRs by treating the metal electrodes as standard anelastic solids. The experimental results shown therein also validate that TED in LVRs is mainly caused by metal electrodes.

Recently, Siddiqi et al. [50] applied Ardito’s formulation from [91] to a series of biconvex AlN-on-Si resonators which suffered more from TED rather than anchor loss. Figure 5a shows the computed thermal profile for a biconvex AlN-on-Si resonator under investigation, illustrating that the temperature gradients occur alternatively along the width of the resonator. This temperature distribution pattern agrees with the S_0_ vibration mode of Lamb wave indicating that these temperature variations are caused by local compression and tension. Due to the presence of thermal gradients, irreversible heat flow arises resulting in TED. Figure 5b shows that the maximum changes in temperature occur at the electrode area which agrees with the analysis results shown in [51]. However, it is shown in [50] that the computed values of *Q_TED_* from Al electrodes alone are about an order of magnitude larger than the experimental values of *Q*. Instead, by using the numerical analysis model, the computed *Q_TED_* as a function of thermal expansion coefficients of AlN layer (α_AlN_) is obtained as shown in Figure 5d, from which it is suspected that the AlN layer has a greater contribution to TED compared to the Al electrodes in AlN-on-Si resonators.

The key challenge to experimentally investigate TED in MEMS resonators is that TED is typically present together with anchor loss, which makes it difficult to analyze TED separately. Thus, it is important to find an approach to separate these two dissipation mechanisms. It is reported by Li et al. [92] that the measured *Q* was increased by about 2.5 times when operating a 61-MHz polysilicon wine-glass disk micro-resonators at cryogenic temperatures down to 5 K compared to room temperature. This experimental result suggests that at least one of the dominant dissipation mechanisms in silicon micro-resonators is dependent on temperature. This dependence of *Q* on temperature were also observed in LVRs using different piezoelectric materials [33,93]. Segovia-Fernandez et al. [31] investigated the effect of temperature on *Q* in a 1-GHz AlN LVR and attributed the observed temperature-dependence of *Q* to TED.

Recently, Tu et al. [52] utilized the cryogenic cooling as a practical approach to analyze the dominance of TED in AlN-on-Si LVRs with different physical geometries and vibration modes. Figure 6a,b shows the optical micrograph and the schematic of the rectangular LVR under investigation, respectively. Two resonators were designed with different supporting tether lengths (Design A: L_a_ = 10 μm; Design B: L_a_ = 45 μm). Figure 6c depicts the symmetrical and anti-symmetrical vibration modes present in both designs. It was experimentally found that the measured values of unloaded quality factor (*Q_u_*) at room temperature for both symmetrical and anti-symmetrical vibration modes in Design A were substantially larger than that of Design B. To clarify the dominant dissipation mechanisms in these two designs with only difference in supporting tether length, cryogenic cooling was used as an experimental investigation means. Figure 6d shows the measured electrical transmission magnitudes (*S*_21_) for the symmetrical mode as one device of Design A is cooled from room temperature to 78 K. It can be seen that operating the device at 78 K doubles the measured *Q_u_* compared to that at room temperature (298 K), and the insertion loss are accordingly reduced by 4 dB. Same cryogenic cooling experiment was repeated over three die samples and consistent results were obtained as shown in Figure 6e, where a linear relationship between *Q_u_* and 1/T can be observed. By contrast, *Q_u_* remained almost constant for devices of Design B upon the same experiment. This suggests that *Q_u_* of Design B is set by anchor loss which is temperature-independent. For Design A, TED plays a more significant role in setting *Q_u_* at room temperature, and TED become smaller as temperature were lowered. These experimental results taken as a whole demonstrate that cryogenic cooling can be used as an effective approach to study the dominance of respective damping sources in LVRs.

## 4. Q-Enhancement Strategies for LVRs

Over the last decade, numerous Q-enhancement strategies have been proposed to reduce either anchor loss or TED, which are widely considered as the two of the most dominant dissipation mechanisms in LVRs. This section will review some of the Q-enhancement strategies.

### 4.1. Q-Enhancement Strategies for Reducing Anchor Loss

Up to now, most of the reported strategies on reducing anchor loss share the same rationale which is to confine the acoustic energy inside the resonator body and prevent it from leaking to the surrounding substrate. To this end, acoustic reflectors, which serve to reflect acoustic energy, are commonly introduced along the acoustic wave travelling path from the resonator body to the substrate. The acoustic reflectors can be either placed at the substrate area or supporting tethers. An alternative method to confine the acoustic energy inside the resonator body is to modify the geometry of the resonator body directly. Some of the work based on these Q-enhancement strategies are reviewed as follows.

Harrington et al. [94] demonstrated a type of arc-shape acoustic reflectors in AlN-on-Si LVRs. The introduction of etched arcs creates a mismatch in acoustic impedances, which is utilized to reflect the acoustic energy back to the resonator body. It is reported in [94] that the acoustic reflector should be positioned at half a wavelength to the resonator in order to obtain largest *Q* enhancement. Recently, Hakhamanesh et al. [95] further developed this Q-enhancement strategy in a 86-MHz AlN-on-Si LVR by introducing the etched notches with specific angles, which is believed to provide smooth acoustic impedance transformation and thus enable higher Q-enhancement. Figure 7a depicts the SEM of the device, where it can be seen that the arcs and notches are etched through the substrate. It was shown that the measured *Q* (19,700) of the LVR was 13.1 times that without arc-shape reflectors and notches (1500) [95]. Instead of etching voids (i.e., subtractive process), the acoustic reflectors can also be realized by additive layers. Gao et al. [96] demonstrated an acoustic reflector by adding a Pt layer regionally and lithographically setting the release area in a 250-MHz AlN LVR as shown in Figure 7b. The reported *Q* enhancement of this method is 3.1 times. Another commonly adopted approach to form an acoustic reflector is to use phononic crystals (PnCs) which feature frequency-defined acoustic band gaps prohibiting propagation of acoustic waves. To serve as an effective acoustic reflector, the band gap of the PnC should be engineered to cover the frequency of the outgoing acoustic wave. Zhu et al. [97] demonstrated this Q-enhancement strategy in a 143-MHz AlN-on-Si LVR, as shown in Figure 7c, where it can be seen that a two-dimensional (2D) PnC array is formed using periodically arranged air-holes as unit cell. The measured *Q* of this device is 3620, which is twice that of the one without employing PnC. Apart from air-holes [98], other shapes of PnC unit cell, such as rings [99], crosses [100], fractals [101], snowflakes [102] and solid-disks [103] were also reported. Among them, it is shown by Ardito et al. [103] that solid-disk PnC unit cell outperforms others in that is has larger band gap width, which is believed to help confine the acoustic waves. Recently, Siddiqi et al. [104] demonstrated this concept in a 141-MHz AlN-on-Si LVR as shown in Figure 7e. In this work, the solid-disk PnC is characterized by a wide bandgap of 120 MHz around a center frequency of 144.7 MHz. The measured enhancement in *Q* is 4.2 times to realize Qs of about 10,000, which is significantly higher than previous works, as shown in Figure 7d. This greater *Q* enhancement confirms that wider bandgap of PnCs helps reduce anchor loss in LVRs. Another concern in designing PnCs is the input/output (I/O) track routing. Since the width of inter-cell link should be as narrow as possible for wider band gap, it becomes impractical to route the I/O track through these narrow inter-cell links. This is the reason why some buffering spaces are left between the resonator and 2D PnC array for I/O track routing as shown in Figure 7c,d. However, this buffer space provides a leakage path for acoustic waves. To address this deficiency, Siddiqi et al. [105] proposed a hybrid PnC array formed by two types of PnC unit cells. Figure 7f shows the SEM of the resonator with solid-disk and ring shape PnC arrays at two sides. The ring-shape PnC array serves as an acoustic reflector while also providing wider links for ease of I/O tracking. The enhancement in *Q* is reported to be 3.7 times which is comparable to that of the PnC design shown in Figure 7e. Instead of placing the acoustic reflectors in the substrate area, it is also feasible to integrate the reflectors on the supporting tethers. Sorenson et al. [106] demonstrated a *Q* enhancement of 1.8 times in a 213-MHz AlN-on-Si LVR by placing acoustic reflectors on supporting tethers. The device is shown in Figure 7g, where it can be seen that a ring-shape 1D array of PnC is used to form the acoustic reflectors. In Sorenson’s work, multiple ring-shape 1D arrays were also used to form the supporting tethers in a 604-MHz AlN-on-Si LVR, which exhibited a 1.6× enhancement in *Q*. Another example of a 1D PnC is the cross-shape 1D PnC by Lin et al. [107] applied to a 555-MHz AlN LVR as shown in Figure 7h. The reported *Q* enhancement was 1.5 times. More recently, Rawat et al. [108] proposed novel asymmetric PnC tethers for reduction of anchor loss in a 313-MHz AlN-on-Si LVR as shown in Figure 7i. It was reported that the device with asymmetric PnC on the tethers exhibited a measured *Q* about 24 times that of using simple beam tethers (the reference resonators based on simple beam tethers had an unusually low Q of around 300). Compared to placing acoustic reflectors in the substrate area, integrating the reflectors directly on tethers favors a more compact size. However, the long and narrow 1D PnC tethers normally make the device vulnerable to mechanical shock [97].

Apart from employing acoustic reflectors in the substrate area and supporting tethers, researchers also resort to more compact design strategies which aim to confine the acoustic energy by making modifications directly on the resonator body. Lin et al. [109] proposed a *Q* enhancement strategy which utilized biconvex edges instead of conventional flat edges. Figure 8a depicts the schematic and the simulated mode shape of such device. It can be seen that from the simulated mode shape that biconvex edges serve to confine the acoustic energy to the central area of the resonator body and thus prevent the acoustic energy from leaking out through the supporting tethers. It is reported that 2.6× enhancement in *Q* was obtained in a 492-MHz AlN LVR. This Q-enhancement strategy was also demonstrated in a series of AlN-on-Si LVRs with resonant frequencies from 70 MHz to 140 MHz [110,111]. A systematic study on the key design parameters of the biconvex LVRs such as mode order, resonator length, curvature and electrode coverage was performed by Siddiqi et al. [111]. It has been experimentally demonstrated in Siddiqi’s work that the level of *Q* enhancement is a function of the curvature of biconvex edges, and the anchor loss can be reduced to a point where TED becomes observable. Figure 8b shows one of such LVR which yields 3.8× enhancement in *Q* compared to flat-edge device. This energy localization phenomenon was analyzed using dispersion theory described by Tabrizian et al. [112], who shows that transformation of propagating waves in the central region to evanescent waves at the flanks occurs when the cross-sectional dimensions of the acoustic waveguide changes. This indicates that the physical dimensions of the flanks no longer impact the intended vibration mode if only evanescent waves are present at the flanks. This idea has led to an interesting work by Ghatge et al. [113,114], which adopted wide tethers to support LVRs instead of using narrow tethers as shown in Figure 8c. The benefits of employing wide tethers include enhancement of power handling capability, less vulnerability to mechanical shock and convenience in routing the I/O electrodes. Instead of designing gradually-changed width in the acoustic waveguides, Zou et al. [115,116] and Lin et al. [117] demonstrated *Q* enhancements in AlN LVRs using more abrupt variations in geometry at two ends of the resonator body such as design with beveled, rounded and chamfered corners as shown in Figure 8d–f. Among these designs, the LVR design using rounded corner exhibited largest *Q* enhancement of 1.6 times. Some design strategies based on local etching were also proposed to reduce anchor loss such as introducing etched slots [118] or etched holes [119] as shown in Figure 8g,h. Apart from the strategies mentioned above, making modification to IDT electrodes were also proposed as an effective means to suppress spurious resonances and thus enhance *Q* [120,121]. Giovannini et al. [120] demonstrated a 1.2× enhancement of *Q* in a 889-MHz AlN LVR using apodized electrode configuration as shown in Figure 8i. This work shows that elimination of spurious resonances helps increase *Q* by focusing more acoustic energy into the intended resonant peak. However, the apodized electrode configuration usually suffers from reduction of *k_eff_*^2^. To boost *Q* without degrading *k_eff_*^2^, Zhou et al. [121] recently proposed a spurious resonance suppression technique for AlN LVR using IDT electrodes with hammerhead structure at the end. This idea borrows from the design of a Piston mode structure, which has been widely used to suppress transverse modes in SAW and FBAR devices [122,123]. Table 1 provides a summary of various Q-enhancement strategies for reducing anchor loss in LVRs.

### 4.2. Q-Enhancement Strategies for Reducing TED

While much work has been done to reduce anchor loss, there has been comparatively far less work reported on suppressing TED. This is partly due to the fact that the important role of TED in setting *Q* of LVRs was realized rather recently [50,51,111]. Given the dependence of TED on metal electrodes, a straightforward strategy to reduce TED is to decrease of electrode coverage in LVRs. Segovia-Fernandez et al. [51] studied a 1-GHz AlN LVR using Au as top electrode and Pt as bottom electrode, and showed that the measured *Q_u_* was doubled when electrode coverage ratio drop from 75% to 25%. This *Q* enhancement by reducing the electrode coverage was consistently observed when Pt and Al were adopted as the top electrode materials. It was also experimentally demonstrated by Siddiqi et al. [111] in a series of biconvex 142-MHz AlN-on-Si LVRs that the measured *Q_u_* can be increased by 60~70% when reducing the number of IDT electrodes from 5 to 3. In comparison, no significant effect in measured *Q_u_* can be observed for conventional flat-edge LVRs, which indicated that the dependence of TED on electrode coverage was more notable as anchor loss was minimized. However, reducing the electrode coverage inevitably increases the motional resistance of LVRs. Apart from reducing the electrode coverage, it is possible to suppress TED by choosing metal electrode materials with optimal thermal characteristics [51]. However, complexity arises in that there are many other factors such as resistivity, acoustic impedance and power durability that need to be considered when choosing the electrode materials. In addition, Yan et al. [56] demonstrated a 940-MHz AlN LVR with non-contacting IDT electrodes, which exhibited 5× enhancement in *Q* compared to the conventional LVR using contacting IDT electrodes. This novel design separates the piezoelectric film from its electrodes by sub-micron gaps, which eliminates the mechanical coupling between the piezoelectric film and the electrodes while keeps the electrical coupling still effective. Although not clarified in [56], it is reasonable to attribute the observed *Q* enhancement to reduction of TED since the electrodes no longer displace after physical separation with piezoelectric film. The problem associated with this non-contacting electrode design is the complex fabrication process as well as the considerable drop in the *k_eff_*^2^ due to the air gap.

## 5. Discussion

### 5.1. Development for Dissipation Analysis Methods

Although some numerical and experimental analysis methods have been proposed for studying anchor loss and TED in LVRs as detailed in Section 3, this research field is still in its infant stage. For numerical analysis methods, the validation of computed values of *Q_anchor_* and *Q_TED_* depends highly on the measured values of *Qs* that are usually very sensitive to fabrication tolerance, especially for the case where high-Q resonators are concerned. Given expected device-to-device variations for *Q* in actual devices, acquiring reliable experimental data to validate models requires at the very least a pool of measured data from which mean values and variances can be obtained [50,51]. It is noted that the research community has not yet reached a consensus on conclusive numerical methods for computing *Q_anchor_* and *Q_TED_* as much work still needs to be done in further validation of each proposed numerical method on a larger test pool of devices. For experimental analysis methods, cryogenic cooling and out-of-plane displacement profiling methods are effective in clarifying the dominant role of respective dissipation mechanism (i.e., anchor loss or TED) in LVRs. However, both experimental analysis methods currently only yield qualitative results. Further work is required to exploit these methods to provide accurate predictions of *Q_anchor_* and *Q_TED_*. In addition, this paper focuses on analyzing anchor loss and TED, which are two dominant dissipations that set the *Q* in LVRs operating in air while having characteristic dimensions at the scale of microns. When operating LVRs in liquid medium or reducing its critical dimensions to the nano-scale, it can be envisaged that other sources of damping such as viscous loss or surface loss would play more dominant roles than anchor loss or TED. As such, numerical and experimental analysis methods targeting other dissipation mechanisms are also highly relevant in this developing field.

### 5.2. Comparison of Different Q-Enhancement Strategies

To suppress anchor loss, acoustic reflectors are either placed at the substrate area or realized as the supporting tethers of LVRs. The former unavoidably increases the footprint of the device. The latter combines the acoustic reflector together with the function of mechanical support achieving a space-efficient solution. However, building the delicate structures required for the acoustic reflector on tethers usually compromise the robustness of the mechanical support, making it vulnerable to mechanical shock. Among these strategies based on acoustic reflectors, PnCs have been extensively researched during the past decade [97,100,104,105]. The obvious advantage of using PnCs as acoustic reflectors is that it usually offers a frequency stopband that prevents the propagation of elastic wave in any type (i.e., longitudinal or shear wave). In comparison, the arc-shape acoustic reflectors placed at half-wavelength distance from the resonator is only aimed at reflecting the longitudinal wave in a narrow frequency band, which falls short of PnCs in terms of reflecting efficiency. One problem for acoustic reflectors based on PnCs is that their stopband characteristics such as center frequency and bandwidth highly depend on the dimensions of the unit cell structure. Generally, smaller dimensions are required for higher center frequency or larger bandwidth, which increases the complexity of the microfabrication process.

An alternative strategy to reduce anchor loss is to modify the resonator body itself. This strategy relies on the efficient confinement of acoustic energy within the resonator body by designing appropriate mechanical boundaries or electrical excitation configurations. The biconvex designs [109,110,111] and the designs with gradual change in width [113,114] all share the same rationale which serves to transform the propagating waves in the central region to evanescent waves at the flanks in order to reduce the leakage of acoustic energy. This strategy has been demonstrated to work across a range of frequencies. However, one problem of this strategy is that sufficient area is required for effective transformation of propagating waves to evanescent waves, which essentially reduces the maximum piezoelectric transduction area achievable [111,114]. There are some other design strategies, which are based on more abrupt geometric modifications, proposed to reduce anchor loss, such as using beveled, rounded or chamfered corners at the ends of the resonator bodies [115,116,117] and introducing etched slots or holes [118,119], but the broad applicability of these methods to different operating frequencies has not been investigated to date. Another strategy to enhance *Q* involves designing the electrical excitation configuration to suppress spurious modes which could carry the acoustic energy away from the resonator body [120,121]. This Q-enhancement strategy is attractive for LVRs using more anisotropic piezoelectric materials as the resonator body. For instance, it has been reported that LVRs based on single crystal Lithium Niobate (sc-LN) film suffer from vulnerability to spurious modes [124].

### 5.3. Q-Enhancement Strategies for Novel LVRs

This paper mainly focuses on the work reported for LVRs utilizing polycrystalline AlN thin film and operating in the S_0_ mode of Lamb wave. Among the different Lamb waves, the S_0_ mode in AlN or sc-LN thin films has received the widest attention due to its suitable acoustic properties such as moderate phase velocity and low dispersion. However, attaining large *k_eff_*^2^ at several GHz, which is highly desired for the application of wide-band RF filters, is difficult for S_0_ mode. Recently, sc-LN LVRs utilizing the first-order asymmetric modes (A_1_) have gained considerable attention due to its high acoustic velocity (>10,000 m/s) and large electromechanical coupling (>10%) [125,126]. It has been demonstrated that sc-LN LVRs operating in A_1_ mode can achieve resonant frequency as high as 5 GHz with *k_eff_*^2^ of 25% [127,128]. However, the reported *Qs* of sc-LN LVRs remain comparatively low relative to AlN LVRs. Although the dominant damping mechanism in A_1_-mode sc-LN LVRs remains unclear, it was found the *Q* can be improved by suppressing the spurious modes, which was realized by introducing etched windows between the electrodes and bus-bar, as well as proper design of the electrodes [125]. This Q-enhancement strategy essentially relies on optimizing the mechanical boundary and electrical excitation configuration.

Another novel type of LVRs that has been heavily researched recently involves the use of Sc-doped AlN (ScAlN) piezoelectric films aimed at enhancing *k_eff_*^2^. Although it has been reported that the *k_eff_*^2^ can be increased by about four times compared to un-doped AlN for a doping concentration of 40%, the improvement comes at the cost of significant reduction in *Q* [129]. Possible reasons for degradation in *Q* are related to the formation of inclusions and film stress during the film deposition process [130]. Besides optimizing the process parameters to avoid inclusions, as well as controlling film stress, an interesting Q-enhancement strategy in terms of materials advances has been to combines the ScAlN film with single crystal silicon to realize high-Q ScAlN-on-Si TPoS LVRs [131].

## 6. Conclusions

Over the past two decades, substantial progress has been made in the research field of piezoelectric MEMS resonators, especially for LVRs. Piezoelectric LVRs feature wide operation frequency coverage (from tens of MHz to several GHz) and provide a pathway towards realizing a multi-frequency integrated solution on a single chip. Significant research effort has been devoted to applying LVRs to realize high-performance resonant sensors, RF filters and oscillators. Among all these applications, LVRs with high *Q* are always desired, spurring great interest in elucidating and modeling dominant damping mechanisms in LVRs, based on which appropriate and targeted Q-enhancement strategies can be formulated and validated. This paper has focused on reviewing the analysis methods for anchor loss and TED, which are known to dominate damping and limit Q in most of LVRs. Various Q-enhancement strategies targeting reducing these two types of losses have also been reviewed and discussed here. Current outstanding issues and potential research directions are highlighted. In particular, in the case of LVRs based on sc-LN and ScAlN thin films that have recently garnered much interest, much work still needs to be done to clarify and understand their underlying dominant sources of damping that set *Q*. We foresee that effective dissipation analysis methods and Q-enhancement strategies for these novel LVRs will be in high demand given their highly promising applications in 5G RF filters.

## Figures and Tables

**Figure 1 sensors-20-04978-f001:**
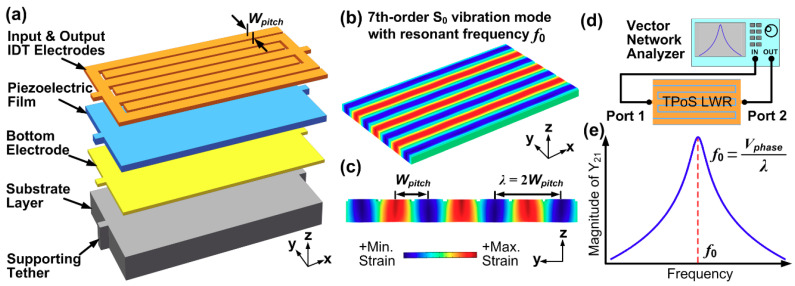
(**a**) Schematic for typical structure of thin-film piezoelectric-on-substrate (TPoS) resonators; (**b**,**c**) perspective and cross-sectional views of the vibration mode shape of 7th-order fundamental symmetrical (S_0_) mode associated with resonant frequency *f*_0_; (**d**) electrical characterization setup for a 2-port TPoS laterally vibrating resonators (LVR); (**e**) typical measured magnitude of transfer admittance *Y*_21_ of a 2-port TPoS LVR.

**Figure 2 sensors-20-04978-f002:**
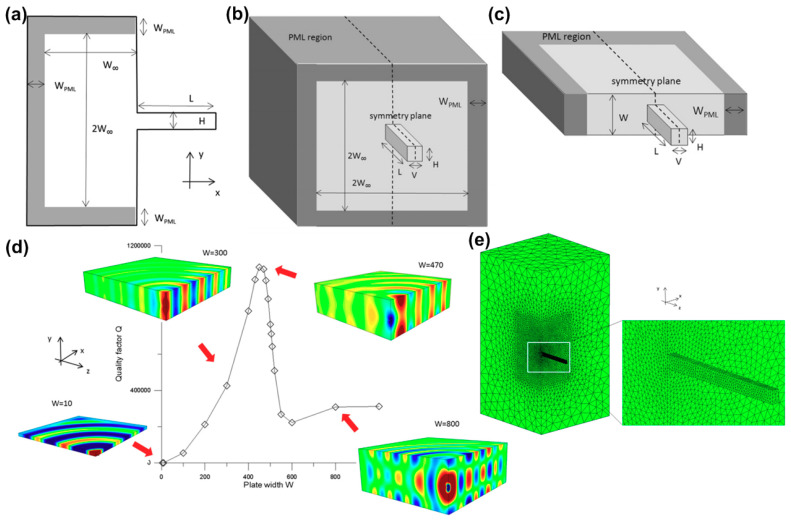
Evaluation of anchor loss in a cantilever structure using numerical analysis: (**a**) implementation of perfectly matched layers (PML) for a 2D cantilever; (**b**) implementation of PML for a 3D cantilever with a semi-infinite substrate; (**c**) implementation of PML for a 3D cantilever with a plate substrate; (**d**) computed values of *Q_anchor_*; (**e**) mesh adopted for the 3D cantilever and semi-infinite substrate. This figure is reproduced from [80] (© 2013 Elsevier).

**Figure 3 sensors-20-04978-f003:**
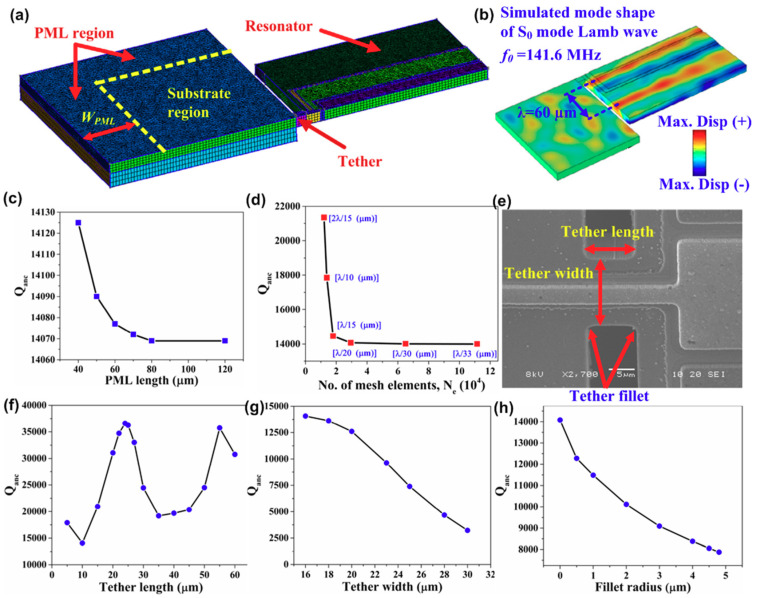
Evaluation of anchor loss in a AlN-on-Si resonator using numerical analysis: (**a**) implementation of PML in a 3D model; (**b**) simulated vibration mode shape of S_0_ Lamb wave at 141.6 MHz; (**c**,**d**) computed *Q_anchor_* in relation to PML length and number of mesh elements; (**e**) scanning electron micrograph (SEM) for the tether part of the fabricated resonator; (**f**–**h**) computed *Q_anchor_* with respect to tether length, tether width and fillet radius. This figure is reproduced from [50] (© 2019 IOP Publishing).

**Figure 4 sensors-20-04978-f004:**
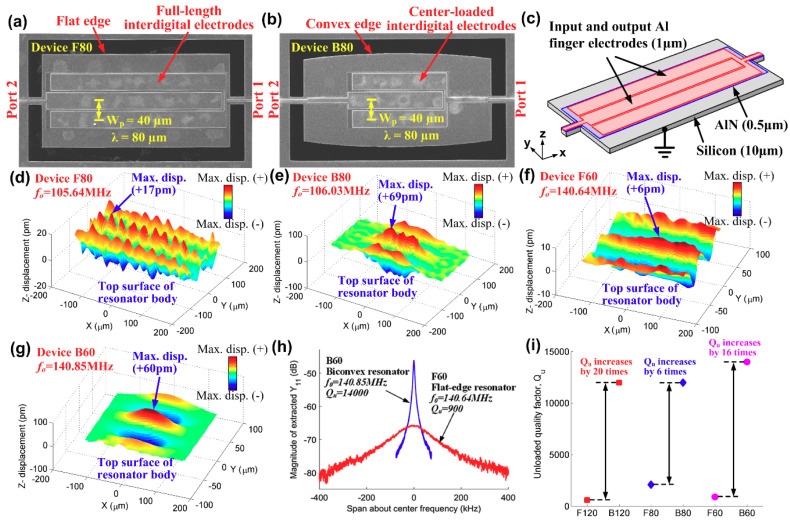
Experimental investigation of anchor loss using laser Doppler vibrometry: (**a**) scanning electron micrograph (SEM) of a flat-edge AlN-on-Si resonator; (**b**) SEM of a biconvex AlN-on-Si resonator; (**c**) schematic of a typical structure for AlN-on-Si LVR; (**d**–**g**) measured z-direction displacement profile on the top surface of four resonators (F80/B80 denotes flat-edge/biconvex resonator with acoustic wavelength of 80 μm); (**h**) measured magnitude of admittance frequency response for flat-edge resonator (F60) and biconvex resonator (B60), showing much higher *Q* of the latter; (**i**) comparison of measured *Qs* among three groups of device with different resonant frequencies (each group includes a flat-edge resonator and a biconvex resonator, which were designed to have similar resonant frequencies but starkly different *Q*). This figure is reproduced from [83] (© 2017 Elsevier).

**Figure 5 sensors-20-04978-f005:**
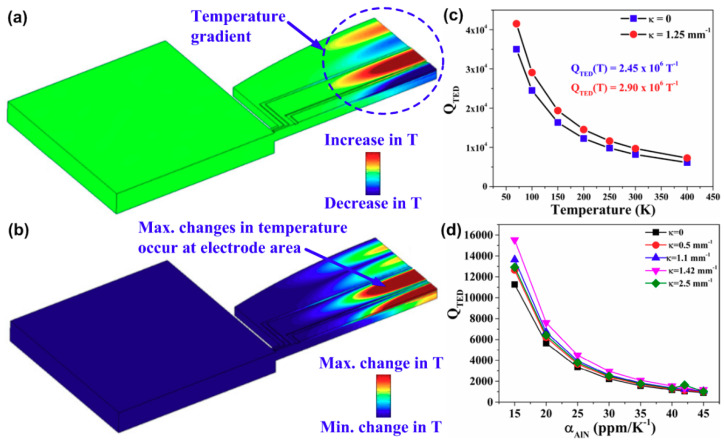
Evaluation of thermoelastic damping (TED) using numerical analysis: (**a**) computed thermal profile showing temperature gradients as a result of local compression and tension; (**b**) computed thermal profile showing that maximum change in temperature occurs at electrodes; (**c**) computed *Q_TED_* for both flat-edge and biconvex resonators as a function of temperature; (**d**) computed *Q_TED_* as a function of thermal expansion coefficient of AlN layer (α_AlN_) for resonators with different curvatures (*κ*). This figure is reproduced from [50] (© 2019 IOP Publishing).

**Figure 6 sensors-20-04978-f006:**
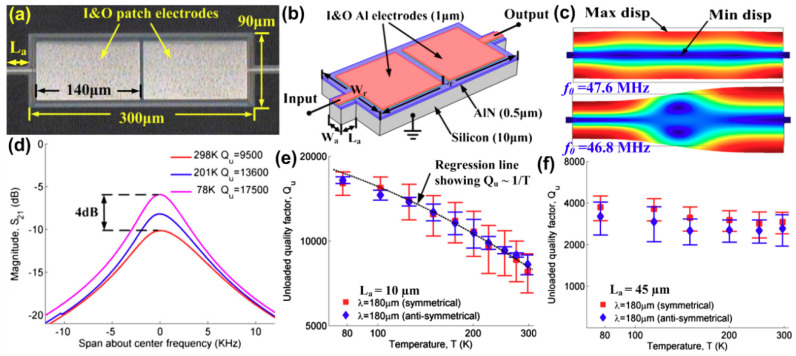
Experimental investigation of thermoelastic damping (TED) using cryogenic cooling: (**a**) optical micrograph of a rectangular Al-on-Si LVR with supporting tethers (length is L_a_) at two sides; (**b**) schematic of the resonator structure showing constituent layers; (**c**) simulated symmetrical and anti-symmetrical vibration modes at 47.6 MHz and 46.8 MHz, respectively; (**d**) measured electrical transmission magnitudes (*S*_21_) for the symmetrical mode as the device is cooled from room temperature to 78 K; (**e**) mean values and standard deviations of measured *Q* for the resonator with L_a_ = 10 μm when cooled from 300 K to 78 K; (**f**) mean values of measured *Q* for the resonator with L_a_ = 45 μm when cooled from 300 K to 78 K. This figure is reproduced from [52] (© 2016 Elsevier).

**Figure 7 sensors-20-04978-f007:**
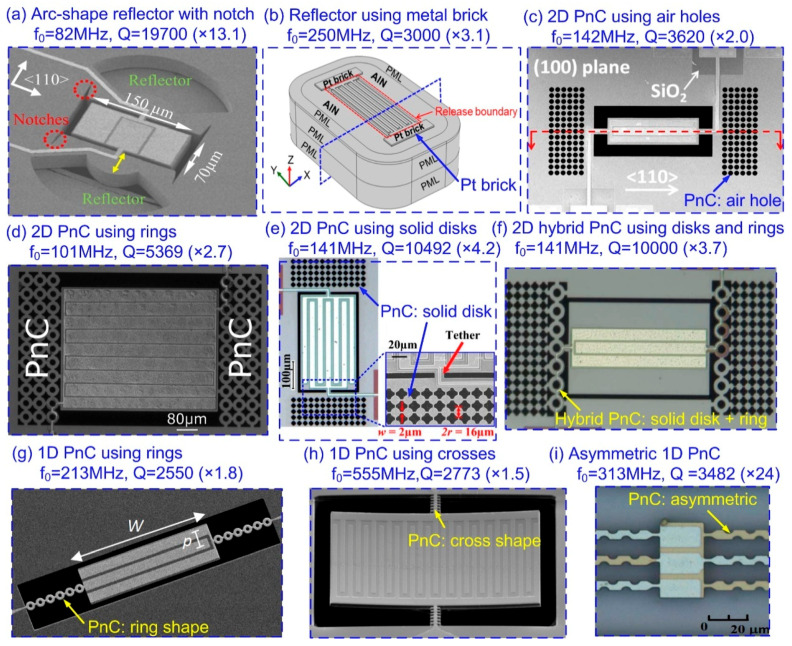
Q-enhancement strategies by placing acoustic reflectors on substrate or supporting tether: (**a**) SEM of an 82-MHz Al-on-Si LVR using etched arcs as acoustic reflectors together with etched notches. This figure is reproduced from [95] (© 2018 IEEE); (**b**) schematic of a 250-MHz AlN LVR using Pt bricks as acoustic reflectors. This figure is reproduced from [96] (© 2019 IEEE); (**c**) SEM of a 143-MHz Al-on-Si LVR with 2D phononic crystals (PnC) arrays of air holes. This figure is reproduced from [97] (© 2015 IEEE); (**d**) SEM of an 101-MHz Al-on-Si LVR with 2D PnC arrays of rings. This figure is reproduced from [99] (© 2016 AIP Publishing); (**e**) SEM of a 141-MHz Al-on-Si LVR with 2D PnC arrays of solid-disks. This figure is reproduced from [104] (© 2018 MDPI); (**f**) SEM of a 141-MHz Al-on-Si LVR with hybrid 2D PnC arrays of solid-disks and rings. This figure is reproduced from [105] (© 2019 IEEE); (**g**) SEM of a 213-MHz AlN-on-Si LVR with ring-shape 1D PnC arrays. This figure is reproduced from [106] (© 2011 IEEE); (**h**) SEM of a 555-MHz AlN LVR with cross-shape 1D PnC array. This figure is reproduced from [107] (© 2014 IEEE); (**i**) Optical micrograph of a 313-MHz AlN-on-Si LVR with asymmetric PnC on the tethers. This figure is reproduced from [108] (© 2017 IEEE).

**Figure 8 sensors-20-04978-f008:**
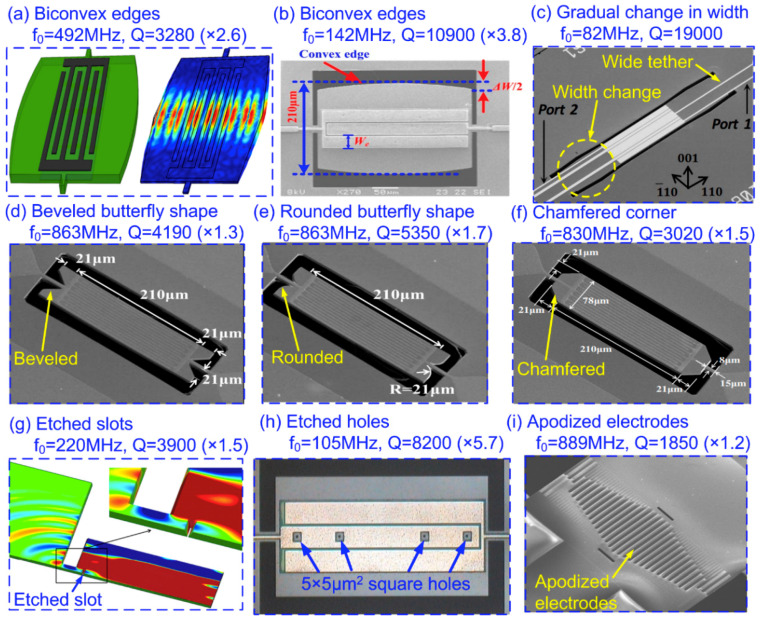
Q-enhancement strategies by geometry modification on the resonator body: (**a**) schematic and simulated mode shape of a 492-MHz AlN LVR using biconvex edges. This figure is reproduced from [109] (© 2011 AIP Publishing); (**b**) SEM of a 142-MHz AlN-on-Si LVR using biconvex edges. This figure is reproduced from [111] (© 2018 IOP Publishing); (**c**) SEM of a 82-MHz AlN-on-Si LVR using gradually-changed width. This figure is reproduced from [113] (© 2019 IEEE); (**d**) SEM of a 863-MHz AlN LVR with beveled butterfly shape; (**e**) SEM of a 863-MHz AlN LVR with rounded butterfly shape. (**d**,**e**) are reproduced from [115] (© 2017 IEEE); (**f**) SEM of a 830-MHz AlN LVR with chamfered corners. This figure is reproduced from [117] (© 2015 IEEE); (**g**) simulated vibration mode shape of a 220-MHz AlN LVR with etched slot. This figure is reproduced from [118] (© 2013 IEEE); (**h**) optical micrograph of a 105-MHz AlN-on-Si LVR with etched holes. This figure is reproduced from [119] (© 2017 Elsevier); (**i**) SEM of a 889-MHz AlN LVR using apodized electrode configuration. This figure is reproduced from [120] (© 2014 Elsevier).

**Table 1 sensors-20-04978-t001:** Summary of Q-enhancement strategies for reducing anchor loss in LVRs.

Acoustic Reflector	Layered Structure	Resonant Frequency	*Q* (Enhanced)	*Q* (Reference)	Increase in Q	Reference
Arc-shape	AlN-on-Si	110 MHz	12042	1818	× 6.6	[94]
Arc-shape	AlN-on-Si	27 MHz	21000	16000	× 1.3	[94]
Arc-shape	AlN-on-Si	86 MHz	19700	1500	× 13.1	[95]
Metal brick	AlN	250 MHz	3000	980	× 3.1	[96]
2D PnC (air hole)	AlN-on-Si	142 MHz	3620	1822	× 2.0	[97]
2D PnC (ring)	AlN-on-Si	101 MHz	5369	2012	× 2.7	[99]
2D PnC (solid disk)	AlN-on-Si	141 MHz	10492	2510	× 4.2	[104]
2D PnC (disk+ring)	AlN-on-Si	141 MHz	10000	2700	× 3.7	[105]
1D PnC (ring)	AlN-on-Si	213 MHz	2550	1400	× 1.8	[106]
1D PnC (ring)	AlN-on-Si	604 MHz	11400	7200	× 1.6	[106]
1D PnC (cross)	AlN	555 MHz	2773	1849	× 1.5	[107]
1D PnC (asymmetric)	AlN	313 MHz	3482	313	× 24	[108]
Biconvex edges	AlN	492 MHz	3280	1255	× 2.6	[109]
	AlN-on-Si	71 MHz	7500	1000	× 7.5	[110]
	AlN-on-Si	106 MHz	7350	1500	× 4.9	[110]
	AlN-on-Si	141 MHz	10894	2866	× 3.8	[111]
Varied width	AlN-on-Si	82 MHz	18955	NA	NA	[113]
Beveled shape	AlN	863 MHz	4189	3348	× 1.3	[115]
Rounded shape	AlN	864 MHz	5352	3348	× 1.6	[115]
Chemfered corner	AlN	829 MHz	3016	2041	× 1.5	[117]
Etched slots	AlN	220 MHz	3902	2360	× 1.7	[118]
Etched holes	AlN-on-Si	105 MHz	8000	1400	× 5.7	[119]
Apodized electrodes	AlN	889 MHz	1849	1585	× 1.2	[120]

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
