# Peer review of "Dissipation Analysis Methods and Q-Enhancement Strategies in Piezoelectric MEMS Laterally Vibrating Resonators: A Review"

_sensors, 2020, doi:10.3390/s20174978_

Round 1

Reviewer 1 Report

The paper reviews the dissipation analysis methods and Q-enhancement strategies in Piezoelectric MEMS Lamb Wave Resonators. While the paper present extensive references on the dissipation analysis method for resonator Q-enhancement, but I am afraid the is fundamental error in the paper, especially on the definition of Lamb wave resonator and which type of MEMS  devices utilise Lamb wave for resonator application.

In line 52, the authors wrote that Lamb wave resonators (LWRs) feature lithography definable resonant frequency that are independent of thickness, which is indeed the main difference between LWRs with FBAR or other flexural beam-based resonators. However, in the examples of LWRs sensor devices in line 59 to 61, the authors mix the example between LWRs devices and other devices based on bulk acoustic waves (BAW) or other flexural beam-based devices. For instant : ref 16 --> contour-mode resonator and ref 17 --> BAW resonator.

Furthermore, in line 63, the authors define two types of LWRs where the first type of LWRs consist of a piezoelectric film sandwich by top and bottom electrodes, using ref 25 as example. This is really a fundamental error because Piazza work in ref 25 is not a type of LWR, but rather a Contour-Mode MEMS Resonators devices. Even thou this type of device is a laterally vibrating devices, but this is not a Lamb waves devices. There are many other works of Piazza that are based on Lamb waves, using both a single thin film material and layered material (piezoelectric and non piezoelectric material).

Due to this fundamental error, I dont think the paper can be published in the current form and require a significant revision, especially if the author want to focus the review on Lamb waves resonator. 

Reviewer 2 Report

Energy dissipation analysis and Q-enhancement in piezoelectric MEMS Lamb waver resonators (LWRs) is a very interesting and nontrivial topic, not only because LWR sensors have a wide range of applications, but also because the dominant energy dissipation mechanism in many piezoelectric MEMS resonators remains an unsolved problem. The author provided a good coverage of anchor loss analysis and corresponding Q-enhancement strategies in LWRs, but as a review paper, the manuscript did not give a comprehensive enough coverage of other dissipation mechanisms in LWRs.

Particularly, electrode-related loss is known to be a major limiting factor in many piezoelectric MEMS resonators and the exact damping mechanism behind the electrode-related loss is unclear in many cases. Various hypotheses have been given in different studies including loss associated with material interface, loss related to film stress, defects in piezoelectric films, and TED contributions from the electrodes. However, there is yet to be a solid universal explanation behind the electrode-related loss in piezoelectric MEMS resonators. The claims of the authors that “TED is the main source responsible for electrode-related loss” and “anchor loss and TED are the most dominant loss mechanisms for LWRs” are inaccurate and misleading. These claims might be true for some special cases, but certainly should not be given as a general claim in a review paper.

Instead of reviewing the different hypotheses behind electrode-related loss in various studies, the authors simply equal it with TED mainly based on ref [37] and [44], none of which provides a convincing support of the statement. The device studied in ref [44] is an AlN resonator where the metal electrode thickness (75-300nm) is comparable with the AlN resonator body thickness (1um). The conclusions drawn in ref [44] cannot and should not be directly applied to the AlN-on-Si resonators reviewed in this manuscript, for which the acoustic energy is largely stored in the Si layer with thickness of 10s of microns. Ref [37] studies AlN-on-Si resonators but it is noted in the paper that TED simulation with standard material properties does not provide results matching experimental observations and a fitted (about 4x larger than normally reported) coefficient of thermal expansion for AlN was proposed to explain the discrepancy. While ref [37] is worth being mentioned in the review paper, I find it unconvincing as a strong support for the authors conclusions made in the review paper.

In order for the manuscript to be accepted, the authors should include more in-depth coverage of recent works on electrode-related and piezoelectric-film-related loss in LWR (or even other types of MEMS resonators, e.g. there have been studies showing electrode coating layer can load the Q of micro-shell resonators unlikely due to TED, T. Nagourney et al. 2015 IEEE ISISS). Such coverage would make the review much more interesting and educational to the readers. The review of TED in LWRs can be kept but should not replace the electrode-related and piezoelectric-film-related loss analysis altogether.

In addition, I have some minor comments:

In introduction line 58-61, besides the applications mentioned by the authors, there have also been recent works reporting inertial sensors realized with piezoelectric LWRs (A. Daruwalla et al. 2020 IEEE Sensors Letters, 2018 IEEE/ION PLANS). These works can also be cited to highlight the recent trend of frequency readout based MEMS sensors.

Equation (4) holds for a single material resonator. For a composite resonator like piezoelectric-on-Si resonator, this equation may not be valid or at least need further modification and explanation.

In section 4.2 line 570, the authors mentioned the tradeoff between electrode-related loss reduction and k_eff^2. For this problem, the work on sidewall-excited LWRs (R. Tabrizian et al. 2011 Transducers) can probably be mentioned as a potential solution.

In section 5 line 590, the author briefly mentioned viscous loss and surface loss. Although they are not the major contributor in many LWRs, they can be critical for certain LWR applications. As a review paper, the authors should try to be more inclusive and these mechanisms worth more than just a couple of sentences. For example, with a strong trend towards NEMS in recently year, more materials reviewing the surface loss in LWR can certainly be interesting for future researchers.

There are a few typos, e.g. line 204 should be “such as acoustic impedance”, line 359 should be “alone” instead of “along”, line 381 should be “…effect of temperature on Q…”, etc.

Round 2

Reviewer 1 Report

I would like to thank the authors for the answer however I have to respectfully disagree with the authors. Nevertheless, probably now I can understand better the source of disagreement.

The authors have provided the definition of Lamb wave and based on that conclude that contour mode resonator in ref [28] can be classified as Lamb wave (also supported by ref [8] that attribute LWR and piezoelectric contour mode resonator as the same class of device). However, I think the authors have misunderstood the difference between Longitudinal bulk acoustic wave (LBAW) and dominantly-longitudinal Lamb Wave (Lamb wave can be classified as a subset of SAW)

The fundamental symmetry mode (S0) Lamb wave at very low plate thickness-to-wavelength (h/λ) ratio has dominantly longitudinal component (close to zero shear component) with a velocity close to LBAW and lowly dispersive. This is the type of LWR that was referred by Yantchev in ref [8]. While this dominantly longitudinal S0 Lamb wave has almost similar characteristic with LBAW, but they are fundamentally different type of acoustic waves. At higher h/λ ratio, the S0 will no longer dominantly longitudinal, with higher shear displacement component and more dispersive.

This h/λ is dependent on the plate thickness h and the wavelength λ. The λ is decided by the period of the interdigital transducer (IDT). For that reason, in ref [8] all examples of LWR are with IDT as a transducer, with and without reflective grating. The LWR devices with IDT and without reflective grating is defined by ref [8] as contour-mode-resonators (CMR)

quote ref [8] : "Accordingly, the term contour-mode-resonators (CMR) is sometimes adopted to specify Lamb wave resonators employing suspended edges". Furthermore, ref [8] refers to this paper below as an example of LWR that is classified as both LWR and CMR

Stephanou, P. J., and A. P. Pisano. "PS-4 GHZ contour extensional mode aluminum nitride MEMS resonators." 2006 IEEE Ultrasonics Symposium. IEEE, 2006.

In both papers, the LWR devices that can be classified as CMR are always using IDT as the transducer, not by electrode sandwich structure in the opposite side of the plate as the authors referred to in ref [28]. Furthermore, in ref [8], figure 1 and figure 2, the difference of FBAR (BAW) and LWR (SAW) is clearly illustrated.

Thus, only a specific type of CMR can be classified as LWR and ref [28] is not a type of CMR that can be classified as LWR. This type of CMR is actually utilised BAW, and more appropriate to classify it as Contour Mode FBAR, for example in the reference below :

Xu, Wencheng, Seokheun Choi, and Junseok Chae. "A contour-mode film bulk acoustic resonator of high quality factor in a liquid environment for biosensing applications." Applied Physics Letters 96.5 (2010): 053703.

Based on the above argument, I think the paper still have a fundamental issue that needs to be addressed before accepting the paper for publication.

Reviewer 2 Report

The revised manuscript is more objective and comprehensive now in terms of analysis of different dissipation mechanisms and I think it is accpetable for publication.

The concerns of the other reviewer regarding LWR defination is reasonable. While I do think the resonators reviewed by the authors fall within LWRs (or at least resonators with Lamb waves, I don't think Lamb wave is limited to SAW and it can exist as BAW with characteristic wavelengths along both plate thickness and plate surface even in a thick plate case), the manuscript certainly does not cover all types of LWRs (e.g. grating type LWR). So some clarifications and justifications of the scope of the review should be added in the manuscript. It could also be helpful to include a brief explanation to clarify the different forms of LWRs and give some examples.

Round 3

Reviewer 1 Report

I would like to thanks the author for the positive responses on the review. I agree that the term of LVR will be more appropiate for the scope of this review paper. Thin film plate acoustic wave resonator (PAW) is a merging of acoustic wave (AW)-based resonator and MEMS resonator, thus it can be approached from two different perspectives, which may caused confusion in the definition. For that reason, I really think it is important to be clear about this. I really think now the review paper is clearer and will be really usefull for bridging the gap from both MEMS and acoustic field. Thus I would like to recommend the paper for publication.